

# Love the one you're with: replicate viral adaptations converge on the same phenotypic change

Craig R. Miller[1,2,3,4], Anna C. Nagel[3], LuAnn Scott[3], Matt Settles[5], Paul Joyce[2,4,†] and Holly A. Wichman[1,2,3]

[1] Center for Modeling Complex Interactions, University of Idaho, Moscow, ID, United States
[2] Institute for Bioinformatics and Evolutionary Studies, University of Idaho, Moscow, ID, United States
[3] Department of Biological Sciences, University of Idaho, Moscow, ID, United States
[4] Department of Mathematics, University of Idaho, Moscow, ID, United States
[5] Bioinformatics Core, University of California, Davis, CA, United States
[†] Deceased.

Corresponding author
Craig R. Miller,
crmiller@uidaho.edu

## ABSTRACT

Parallelism is important because it reveals how inherently stochastic adaptation is. Even as we come to better understand evolutionary forces, stochasticity limits how well we can predict evolutionary outcomes. Here we sought to quantify parallelism and some of its underlying causes by adapting a bacteriophage (ID11) with nine different first-step mutations, each with eight-fold replication, for 100 passages. This was followed by whole-genome sequencing five isolates from each endpoint. A large amount of variation arose—281 mutational events occurred representing 112 unique mutations. At least 41% of the mutations and 77% of the events were adaptive. Within wells, populations generally experienced complex interference dynamics. The genome locations and counts of mutations were highly uneven: mutations were concentrated in two regulatory elements and three genes and, while 103 of the 112 (92%) of the mutations were observed in ≤4 wells, a few mutations arose many times. 91% of the wells and 81% of the isolates had a mutation in the D-promoter. Parallelism was moderate compared to previous experiments with this system. On average, wells shared 27% of their mutations at the DNA level and 38% when the definition of parallel change is expanded to include the same regulatory feature or residue. About half of the parallelism came from D-promoter mutations. Background had a small but significant effect on parallelism. Similarly, an analyses of epistasis between mutations and their ancestral background was significant, but the result was mostly driven by four individual mutations. A second analysis of epistasis focused on de novo mutations revealed that no isolate ever had more than one D-promoter mutation and that 56 of the 65 isolates lacking a D-promoter mutation had a mutation in genes D and/or E. We assayed time to lysis in four of these mutually exclusive mutations (the two most frequent D-promoter and two in gene D) across four genetic backgrounds. In all cases lysis was delayed. We postulate that because host cells were generally rare (i.e., high multiplicity of infection conditions developed), selection favored phage that delayed lysis to better exploit their current host (i.e., 'love the one you're with'). Thus, the vast majority of wells (at least 64 of 68, or 94%) arrived at the same phenotypic solution, but through a variety of genetic changes. We conclude that answering questions about the range of possible adaptive

trajectories, parallelism, and the predictability of evolution requires attention to the many biological levels where the process of adaptation plays out.

## INTRODUCTION

An important question in evolutionary biology is how deterministic and thus potentially predictable, vs. stochastic, and thus less predictable, is the process of adaptation. The answer to this depends on many things that we understand poorly. For example, when a population is not optimally adapted to its environment, how many different phenotypic solutions are available to it? How different are they and how do their fitness peaks compare? For each of these phenotypic solutions, how many mutational pathways are available at the genetic level and how probable are each of these? How do population dynamics influence which solutions win out? Each of these questions is complicated in itself and they become considerably more complex when we consider other facets of reality such as changing environments, frequency-dependent dynamics and interacting species that are also adapting.

Experimental evolution is a venue where we can make inroads into answering these questions. Here we can control many of the confounding variables like the initial genetic conditions, the population size and the biotic and abiotic environments. We can perform experiments in replicate and therefore take a probabilistic view of things. We can archive populations as they adapt so as to preserve a detailed record of the adaptive trajectories each population took. We can also engineer in genetic changes that allow us to study characteristics of routes not taken.

One observation from experimental evolution that relates directly to our opening question about determinism and predictability is that parallel and convergent evolution can be quite common. *Wichman et al. (1999)* allowed two initially identical populations of bacteriophage $\phi$X174 to evolve in chemostats under high temperatures for 10 days. Whole genome sequences of clones from the endpoint populations contained 14 and 15 mutations, seven of which were shared between them. Similarly, *Bull et al.* (*1997*) propagated a total of nine $\phi$X174 lineages under a more complex experimental design and found that of 119 observed substitutions, over half were found in at least two different lineages.

Bacteriophage have very compact genomes and few genes ($\phi$X174 has 11 genes packed in a genome of just over 5 KB) and it is possible that this is a driving force behind the high levels of observed parallelism. Experimental evolution with bacterial systems indicates that while parallelism at the nucleotide level may be rare in more complex organisms, parallel changes are not uncommon at higher levels of biological organization. *Woods et al. (2006)* assessed patterns of parallelism among 12 *Escherichia coli* lineages evolving for 20,000 generations on glucose-limited media (*Lenski et al.*, *1991*) and found that while the pairwise incidence of shared changes at the nucleotide level is quite low (around 2%), it is much higher when we consider mutations in the same gene or operons to be parallel

events. Similarly, *Tenaillon et al. (2012)* adapted *E. coli* to high temperature in replicate and found that while just 2.6% of non-synonymous mutations were shared between lineages, 20% of modified genes and 25% of affected operons were shared. *Chou & Marx (2012)* studied replicate adaptation in an engineered *Methylobacterium extorquens* where the native pathway for metabolizing methanol was replaced by a foreign pathway. Starting from overexpression, they found that all lineages evolved to reduce gene expression, but this was done via three very different mutational pathways (reducing gene copy number, reducing transcript stability and integration of pathway from plasmid to genome). These studies suggest that similar changes at the phenotypic level are sometimes underwritten by changes at the same nucleotide or codon position, sometimes owed to changes in the same gene or operon and sometimes can have very distinct genetic bases.

In this study we assessed how similar the adaptive trajectories are among a set of replicate lineages that begin either as genetically identical or that differ by having different first-step mutations. By performing replicate flask-passage adaptations of the G4-like bacteiophage ID11, *Rokyta et al. (2005)* identified nine first-step beneficial mutations. Here we adapted each of these nine first-step backgrounds under eight-fold replication for 100 passages on the same host, media and temperature, but in microtiter plates rather than in flasks. We then sequenced five clones from each of the 72 lineages and compared genomes to assess patterns of parallel evolution. Similar to the bacterial studies cited above, we found that parallelism is dramatically higher at the phenotypic level than the genetic one.

## MATERIALS AND METHODS

Here we provide summaries of the materials and methods. Substantially greater detail is provided in the Supplemental Information that accompanies this paper.

### Adaptation experiment

ID11 (GenBank accession number AY751298; *Rokyta et al.*, *2005*) is a single-stranded DNA bacteriophage of the family Microviridae. It has a genome of 5,577 bases encoding 11 genes arranged in the same way as G4 (from which it differs by ∼3%). We used the nine first-step beneficial mutations obtained by *Rokyta et al. (2005)* via flask-passaging and adapted each of these genetic backgrounds in eight-fold replicate for 72 total lineages in multiwell plates for 100 passages. Passaging began by adding $10^4$ phage into 500 μL of *E. coli* C host cells at a density of $10^8$/mL. Passaging was done by allowing phage/cell growth for 25 min in flat-bottomed 48-well plates at 37 °C in an incubator shaking at 200 rpm. Plates were then put on ice and 5 μL of each well's volume was transferred to the wells of a fresh plate containing naive hosts in the same volume and density. During growth, wells were sealed with a double-layer of gas permeable membrane to reduce the likelihood of contamination between wells. Five transfers occurred each day and plate were stored overnight in the refrigerator.

### Sampling and sequencing

After 100 transfers we plated and picked five isolates from each lineage. Whole genomes were then sequenced using the Fluidigm Access Array Platform (South San Francisco, CA) and Roche 454 Genome Sequencing FLX (454 Life Sciences, Branford, CT) using

a novel tagging method so that many small genomes could be sequenced at once while retaining linkage information. Raw Roche 454 unclipped DNA sequence reads were cleaned, assigned to barcode and amplicon, and mapped to the ancestral genome using the R package rSFFreader (*R Core Team*, *2014*; *Settles et al.*, *2011*) and a custom R script (See Supplemental Information for more detail). Mutations were accepted as real only when coverage at the site where a mutation was detected was $\geq 10$ and the mutation was present in $>90\%$ of the reads (mean coverage, $46\times$). To avoid strong sample size effects, we removed wells from the analysis with three or fewer successfully sequenced isolates. This resulted in the removal of four wells leaving 68 in the analysis. For each of the 68 wells we then constructed parsimony trees manually to visualize the mutational relationships among the isolates.

## Fitness assay

Because 15 of the 68 accepted wells contained one or more isolates where the ancestral mutation had reverted to wild type, we were led to question our original assumption that all background mutations were beneficial. We therefore conducted a follow-up set of competition fitness assays under conditions that matched the adaptation experiment. In these, we competed each background mutation (plus another mutation that commonly arose, A–G at site 1910, which we denote 1910aG) against wild type ID11 for six passages and estimated fitness from changes in mutation frequency. For each mutation we seeded three wells with a 90:10 mutant to wild type ratio and, in three more wells, the reciprocal ratio of 10:90. Sampling was done at passages 0, 2, 4 and 6 and population sequencing was conducted using the techniques described above except primer concentrations were manipulated to render 24-fold greater coverage in the region of the ancestral mutation.

## Statistical analyses
### Selection coefficients of background mutations

We estimated the selection coefficient ($s$) of each mutation relative to the wildtype by regression. Specifically, we fit the natural log of mutation frequency as a function of generation (assuming two generations per passage) using the glm function in R. The slope of this regression estimates $s$. We then averaged over the six replicate estimates.

### Reversion probability between backgrounds

To test whether background had an effect on the probability of reversion we used a likelihood ratio test (LRT). In short, we calculated the log-likelihood of the data (i.e., probability of observed presence/absence of a reversion in each well) under the null model where the probability is the same across all backgrounds, under the alternative model where each background has a unique probability, and took the difference, $\Lambda$. The $p$-value was determined assuming $\Lambda$ follows a Chi-squared distribution with $df = 8$ (9 backgrounds $- 1$).

### Effect of background on parallelism

We defined parallelism between the pair of wells $i$ and $j$, $P_{ij}$, as the average of two comparisions: the proportion of mutations in $i$ found in $j$ and the reciprocal proportion of $j$ found in $i$. For each background, we calculated $P_{ij}$ for all pairs of wells and

then, by partitioning the sum of squares, calculated how much of the variation in $P_{ij}$ could be attributable to background ($R_{real}$). Because each well was used in multiple comparisons, $P_{ij}$ values were non-independent. We therefore used a data randomization technique described in the Supplemental Information to generate the distribution of $R$ under the null (i.e., no background effect) and from this distribution approximated the $p$-value.

### Epistasis

We analyzed our data for epistasis using three methods. *Method 1*: Continuing with our parallelism theme, one signature of epistasis is for parallelism to be greater within than between backgrounds. Therefore we repeated a similar calculation to that described above, except we calculated $P_{ij}$ between all pairs of wells (i.e., both within and between backgrounds). We then averaged all within-background values ($\bar{P}_{within}$), averaged all between background values ($\bar{P}_{between}$) and took the ratio ($\bar{P}_{w/b(real)} = \bar{P}_{within}/\bar{P}_{between}$). We obtained a $p$-value for the observed ratio using a bootstrap randomization process. *Method 2*: A more powerful test for epistasis can be achieved by asking whether there are nonrandom associations between individual mutations and backgrounds. We did this using a likelihood ratio test where we calculated the difference in the log-likelihood ($\Lambda$) under a null model where a mutation's probability of arising is the same in every well (irrespective of background) and an alternative model where it depends on the background. The $p$-value of $\Lambda$ under the null was again estimated using randomization. This is a global test where one highly non-random association can produce a small $p$-value. To identify which individual mutations show strong epistasis, we removed the mutation with the smallest individual likelihood, reran the entire test, and repeated until the $p$-value was no longer $<5\%$. *Method 3*: We were also interested in whether pairs of de novo mutations show nonrandom associations. To identify these, we first removed all singletons and reversions for which the question is not pertinent. We then took each pairwise combination of mutations and calculated the number of wells in which the pair was observed in one or more isolates. The $p$-value under the null (where the count was due to chance co-occurrence) was determined by randomization. A two-tailed test was performed so both attraction and repulsion would be detected. We used this test to probe for interesting patterns rather than draw firm conclusions and hence did not do a correction for multiple tests. Still, we did remove the most obvious source of false positives by excluding cases where $p < 0.05$ and the combination co-occurred just once.

### Time to lysis assay and analysis

The result from Method 3 in conjunction with evidence in the literature led us to hypothesize that a number of the mutations that arose in the experiment delayed lysis. To test this, we identified four putative lysis-delaying mutations (1910aG, 1911cT, 2131cT and 2134tC) where we had one or more backgrounds both with and without the individual mutation in our set of sequenced isolates. We then performed an assay for the time to lysis where phage were added to cells in shaking flasks at 37 °C, sampled at one minute intervals and plated to determine when titer begins to rise. We then averaged the estimated burst size over replicates for each time point for each genotype and calculated standard errors.

## Quality control

Contamination was a concern in our experiment because adaptations involved 100 transfer events for each of three 48-well plates using a multichannel pipette. We flagged potential contamination events based on four criteria: (1) the expected background mutation was missing from one or more isolates from the well, (2) in those isolates, one of the other background mutations appeared, (3) the putative contamination isolates carried other mutations (besides the other background mutation) that could link them to another well on the same plate and (4) reversions were otherwise rare in wells with the background we expected to find in the well. Upon obtaining our sequence data, one well showed all four red flags and was removed from the analysis. Three other wells met criteria 1 and 2, but not 3 and 4. It was impossible to know if the results from these wells was real evolution or represented contamination. Given the overall scarcity of evidence for contamination we suspect these represent real evolution. We therefore present our analysis with the wells included, but we also reran the analysis without them to confirm that their inclusion or exclusion has no qualitative effect on our results and conclusions.

## RESULTS AND DISCUSSION

### Overview

The goal of this research was to assess parallelism and epistasis during adaptive evolution. The study, however, did not unfold as expected and we therefore begin with an honest summary of what transpired. The experimental design was to begin with nine different first-step beneficial mutations, allow each to further adapt, and have enough replication to formally assess how the accrued mutations depend on initial background, de novo mutations and chance. The nine background mutations, obtained in previous work (*Rokyta et al.*, *2005*), all arose and were highly beneficial in flasks. To achieve the desired replication (8 replicates per background) we shifted from passaging in flasks to sealed microtiter plates and transferred a fixed volume instead of a target number of phage. Titering during the experiment revealed that initial low MOI conditions in the plates rapidly gave way to high MOI ones. Sequencing endpoint isolates then revealed that 15 of the 68 wells (4 wells were removed due to sequencing failures) contained reversions, leading us to question our assumption that the background mutations were beneficial. Subsequent fitness assays and analysis (see Supplemental Information) led us to conclude that these background mutations were adaptive in a low MOI environment but not in a high MOI environment.

We analyzed the data from the perspective of parallel evolution and two major patterns emerged. First, we found that wells converged on the same phenotypic change. Most isolates in most wells, regardless of background, had exactly one mutation in the D-promoter. We further found a small number of other mutations just downstream from the D-promoter (in the coding region of genes D and E) that were mutually exclusive of each other and of the D-promoter mutations. The two most frequently observed D-promoter mutations have also been observed in experimental evolution of the bacteriophage $\phi$X174 (*Wichman et al.*, *1999*; *Wichman, Millstein & Bull*, *2005*) and shown to down-regulate genes D and E (*Brown et al.*, *2010*). This led us to hypothesize that our D-promoter mutations and the

downstream changes in the D/E genes were down-regulating expression or disrupting function of the lysis protein E and thereby delaying lysis. We reasoned that delay of lysis might be favored in the high MOI environment in which we passaged because it could increase burst size with little cost (since subsequent infections are unlikely at high MOI). We conducted an assay with several of the putative E-down-regulating mutations on several different genetic backgrounds and confirmed that they do indeed delay lysis. To quote Stephen Stills (who quoted Billy Preston), *if you can't be with the one you love, love the one you're with* (*Stills*, *1970*).

Second, we found that the prevalence of parallelism depends on how and at what biological level we measure it at. From the population perspective, the proportion of nucleotide mutations shared between independent wells was 28%. If we redefine shared changes to be mutations in the same codon or regulatory feature, this value grows to 38%. Alternatively, we can look at individual mutations and ask how often they occur in more than one well. Overall, 36 of the 112 mutations (32%) were observed in more than one well. This moderate value, however, obscures how uneven the distribution of occurence was: 76 of the 112 mutations (68%) were found uniquely in one well and another 27 (24%) were found in between two and four wells. By contrast, 62 of the 68 wells (91%) we included in the analysis had a mutation in the D-promoter. We show below that the D-promoter mutations as well as a number of mutations just downstream of it delay lysis. Thus, at the phenotypic level, at least 64 of the 68 wells ($\geq$94%) had mutations that delay lysis. Depending on the level of biological organization considered, parallelism might be characterized as low as 28% or over 90%, highlighting the need be attentive to the many biological levels at which the process occurs.

## Mutational dynamics

After quality control (see Supplemental Information) we had 338 sequenced isolates distributed among 68 wells (4.97 isolates per well). Wells on average contained 4.5 mutations (range 2–9, sd = 1.5). We constructed parsimony trees for each well, in part to visualize the data, and in part to assess the types of mutational dynamics within the wells. The trees indicate that complex interference dynamics prevailed during the experiment—as we would expect in large populations—where multiple mutations of similar effect arise, compete and rarely fix before more mutations arise and begin increasing in frequency. Figure 1 illustrates the basic phylogenetic patterns and their prevalence. In just 4 of the 68 wells (6%), all of the sampled isolates were identical; this is consistent with, though not exclusive to, a simple model of sequential selective sweeps (*Gillespie*, *1984*) (Fig. 1A). In the remaining 64 wells (94%) the population was not fixed. Models of adaptation differ with respect to the background(s) upon which secondary beneficial mutations arise. In the classic clonal interference model of *Gerrish & Lenski (1998)*, multiple single mutations may arise and compete on a previously fixed background, but one of these secondary mutations fixes before the process repeats. In our data, 18 of the wells (26%) are consistent with this model (i.e., observations are confined to single tip mutations and their shared background; Fig. 1B). The other wells showed more complex, multiple-mutation dynamics (*Desai & Fisher*, *2007*) where beneficial mutations arise on unfixed backgrounds. In 5 of the wells

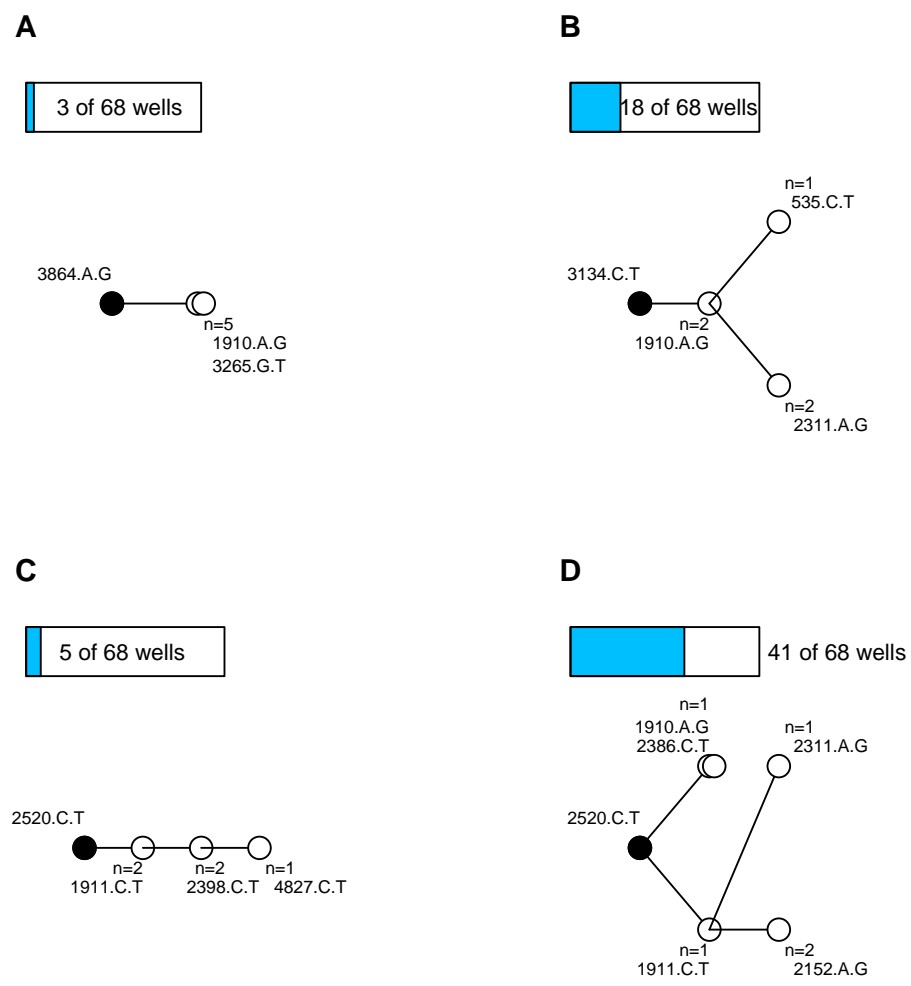

**Figure 1 Most wells displayed complex interference dynamics.** Shown are examples of within-well phylogenies corresponding to different types of mutational dynamics and their relative frequencies. Ancestral background mutation is black dot; de novo mutations are white; bars above trees show number of wells with this basic pattern. (A) All isolates fixed for same mutations is consistent with sweep dynamics. (B) Observations confined to single tip mutations and their shared background consistent with the clonal interference model of *Gerrish & Lenski (1998)*. (C) Unbranched tree in an unfixed population is consistent with mutations arising sequentially on the most-fit background. (D) Extended branching tree in an unfixed population indicates competing multi-step lineages. (C–D) Represent complex interference dynamics.

(7%) the trees were unbranched, consistent with mutations arising sequentially on the single most-fit background (Fig. 1C) while in 41 of the wells (60%) the tree was branched in ways that suggest competing multi-step lineages (Fig. 1D) (*Miller, Joyce & Wichman, 2011*; *Desai, Fisher & Murray, 2007*). The entire set of phylogenies are presented in the Supplemental Information. These findings add to a growing body of evidence (e.g., *Kao & Sherlock, 2008*; *Miller, Joyce & Wichman, 2011*; *Desai, Fisher & Murray, 2007*; *Lee & Marx, 2013*; *Lang et al., 2013*; *Frenkel, Good & Desai, 2014*) that mutational dynamics in large populations usually involve complex interference dynamics.

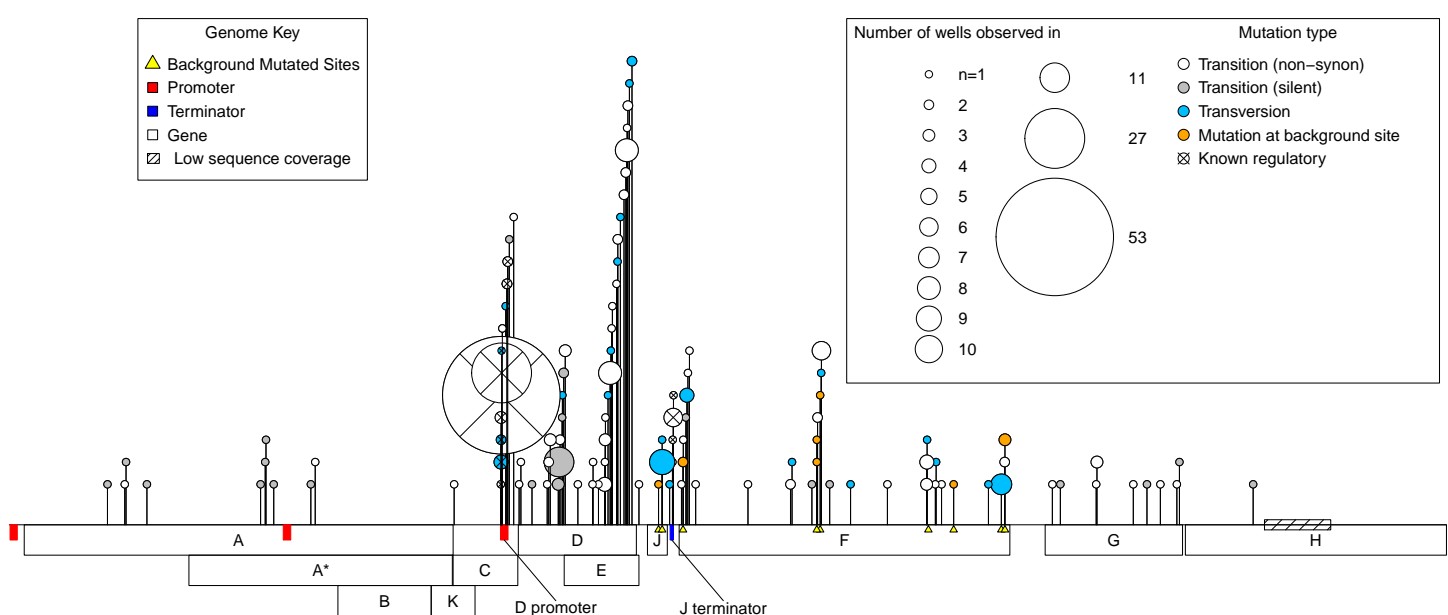

**Figure 2** **Genome location of all mutations observed across the experiment.** A horizontal map of the genome is shown at the bottom of the plot; de novo mutations are indicated with pins above it. The size of the pin head indicates the number of independent wells with the mutation and the pin head color gives information about the mutation itself as described in the inset legend. Gene functions: A, DNA replication; A*, inhibits host DNA replication; B, internal scaffolding protein; C, DNA synthesis; D, external scaffolding protein; E, host cell lysis; F, major coat protein; G, major spike protein; H, pilot protein for DNA injection; J, DNA binding protein; K, unknown.

## Characterizing the mutations

A total of 112 de novo mutations were observed across all wells. These spanned most of the genome and fell within all but one gene. However, mutations were highly uneven both in where and how many independent times (wells) they occurred (Fig. 2). By both of these measures, activity was concentrated in the D-promoter and in genes D and E (which have overlapping reading frames). It was also moderately high in the J terminator and in gene F. Of the 112 mutations, 15 were in regulatory regions or known to have regulatory effects, 76 were non-regulatory and non-synonymous, and the remaining 21 were silent. Multiplying each mutation by the number of wells it occurred in, we have 281 de novo mutational events in the experiment; 95 (33%) of these were mutational events in the D-promoter, 80 (28%) were in genes D and E, 57 (20%) were in gene F and 49 (19%) of the events were elsewhere in the genome. Matrix files with all the information about every observed mutations in every isolate from every well and the ancestral state of that well are available as Supplemental Information.

Neutral variation will tend to reduce parallel evolution. This leads us to ask, how much of the observed variation is adaptive vs. how much is neutral? To answer this, we note that in our data there are two signatures that a mutation is beneficial: (1) when it appears in multiple wells independently, and (2) when it rises to moderate or high frequency unaccompanied by another mutation. The phylogenies from each well contain information about which mutations meet this second criteria as explained in the Supplemental Information (subsection Identifying Neutral vs. Beneficial Mutations). Based on these

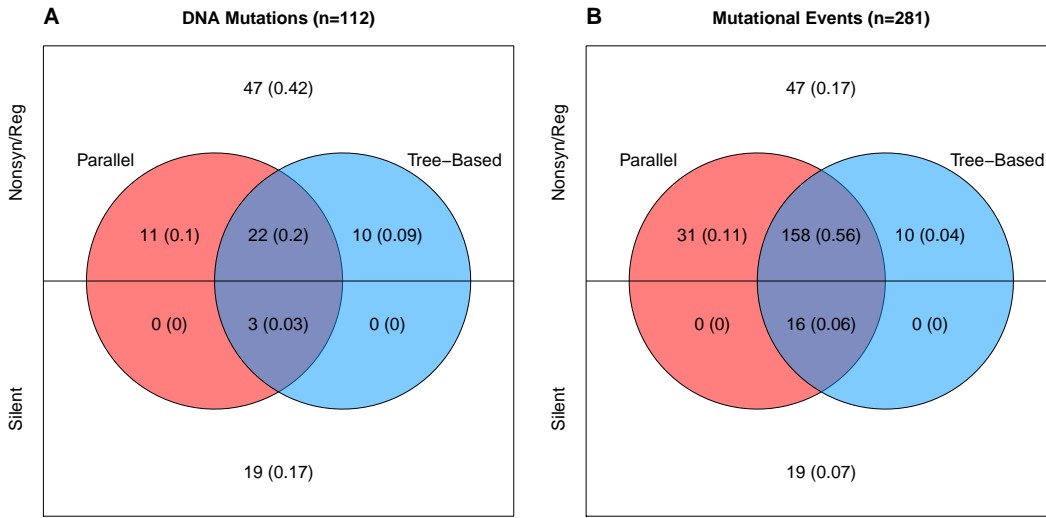

**Figure 3** **Number and evidence of adaptive mutations.** (A) All 112 unique mutations. (B) All 281 mutational events (where the contribution of each mutation is the number wells it appears in). Mutations in colored regions are likely adaptive because they are observed in more than one well independently (red), because one or more well phylogenies implies they rose above low frequency (blue), or both (purple). Mutations in white lack evidence of being adaptive.

lines of evidence, at least 46 of the 112 mutations (41%) are adaptive (Fig. 3A). But because many of these changes occur in multiple wells, the 46 adaptive mutations comprise 215 of the 281 de novo mutational events observed in the experiment. Thus, at least 77% of the mutational events are adaptive (Fig. 3B). The Venn diagram also partitions the mutation and event counts into silent changes on the one side vs. those that are non-synonymous and in regulatory regions (e.g., in the D-promoter) on the other. Two things jump out. First, 41% of the mutations and 17% of the mutational events are both nonsynonymous and lack any evidence of being adaptive. Second, there are only 22 silent mutations in the data. Three of these, corresponding to 16 (or 6%) of mutational events, are adaptive.

## Parallel evolution

The focus of this study is to assess the level of parallel evolution. We begin there, digress into one of its underlying determinants–epistasis–and then follow this topic back around to parallel evolution. Parallel evolution refers to the same change occurring in independent trials of evolution. Although conceptually simple, quantifying it involves some nuance as illustrated by the following three questions. First, what constitutes the same change? We might define shared changes at several biological levels: matching nucleotide change, matching amino acid changes, a change at the same residue, in the same protein domain, in the same gene, in the same regulatory element, part of the same regulatory network, or affecting the same phenotype. Second, should we quantify the frequency of parallelism from the perspective of the mutations or of the replicate populations? In other words, we can ask how often change $x$ co-occurs in two trials, or we can ask of a random pair of replicates, how many or what proportion of the changes are shared. Third, how similar
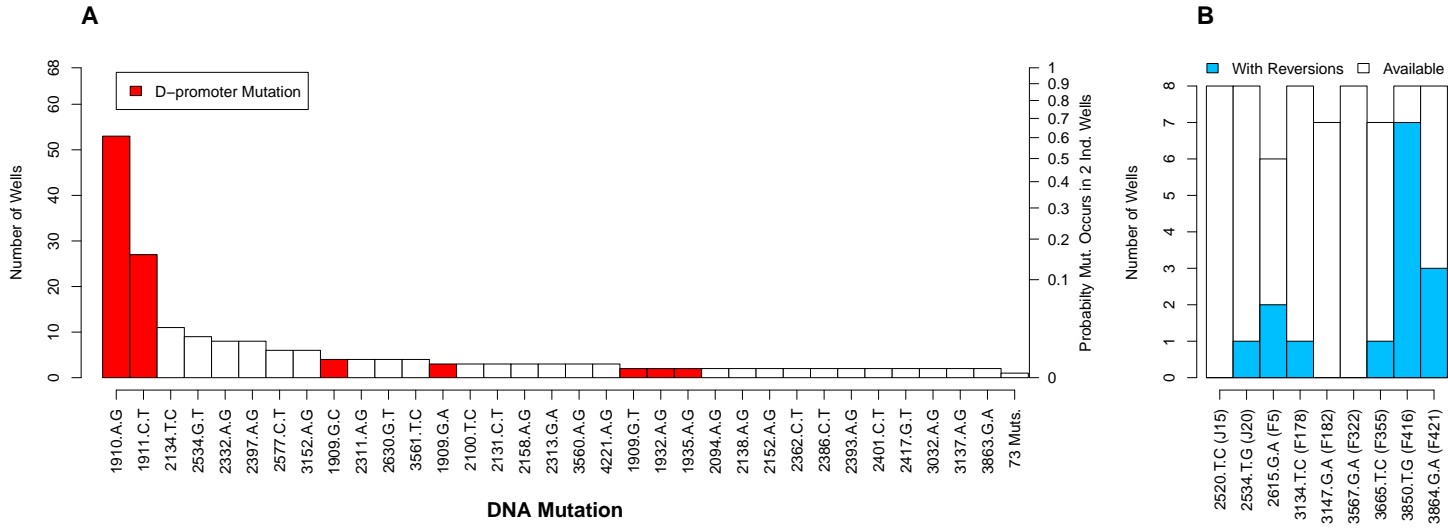

**Figure 4  The number of occurrences of each mutation is highly uneven.** (A) All mutations except reversions. D-promoter mutations observed > once emphasized in red (four more were observed once). *Y*-axis on left gives the well count out of 68; on right is the probability the particular mutation would occur in two wells selected at random (frequency squared). (B) Reversions only. Because 4 wells were dropped from dataset, not all backgrounds had the same number of wells available for reversion (as indicated by white bars). The probability of reversion is not equal across backgrounds ($p = 0.0002$).

must the starting points of the trials be to consider them replicates? One thing we wish to emphasize here is how different the answers can be.

### Parallel evolution by mutation

Of the 112 DNA mutations observed in the experiment, 36 (32%) were observed in more than one well whereas 76 (68%) were observed in only one well. Six of the observed mutations were reversions. We begin by focusing on the non-reversions as this is where the opportunity for evolution is the same across wells. When we plot the distribution of well occurrences by mutation, we find it is highly uneven. Two D-promoter mutations are found often (in 53 and 27 wells) while 103 mutations (92%) are found in four or fewer wells (Fig. 4A). With an observed well frequency of 0.78 (53/68), the probability of finding the D-promoter mutation 1910aG in each of two randomly selected wells is 0.61 ($0.78^2$). By contrast, a mutation like 2397aG that is found 8 times (frequency 0.12) has just a 0.014 probability of co-occurring in a randomly selected pair of wells. When we do this calculation for all 106 mutations and take an unweighted average, we find that a randomly selected mutation in this experiment has only a 0.009 probability of occurring in each of two randomly selected wells.

Although the D-promoter mutations 1910aG and 1911cT arise far more often than other mutations, the distribution in Fig. 4A probably understates the level of parallelism. If we assume that all changes in the D-promoter have similar regulatory effects (an assumption we return to later), then it is appropriate to group them together as a single mutational feature. Upon doing so we find that 62 of the 68 wells (91%) have a D-promoter mutation and that the probability of two random wells each having one is 0.83. We also clustered all

DNA mutations that alter the same codon, but because very few mutations affect the same residue, this had virtually no impact on the distribution of counts.

We now turn to the reversions. Each background had between 6 and 8 replicate opportunities to revert in the experiment. The number of observed reversions differed dramatically by background, ranging from zero in three backgrounds up to seven in background F416 (Fig. 4B). A likelihood ratio test confirms that background has a significant effect on the probability of a reversion ($p = 0.0002$). Thus, for some backgrounds, reversions were a major contributor to parallel evolution and at others they were not. This finding also highlights that genetic background may influence how likely two wells are to share a mutation. We provide to a more detailed analysis of these mutational interactions in the epistasis section below. More immediately, we use this result to justify why, in the next analysis on parallelism between wells, much of our focus is within (not between) backgrounds.

### Parallel evolution by well

In the last section we asked how likely a given mutation was to appear in two independent wells. We now change the conditioning and ask, for two independent wells evolved from the same background, what proportion of the mutations observed in them are shared? The short answer, as we will show in the next paragraph, is 28% at the nucleotide level and 38% when mutations are defined at the regulatory/codon level. At the DNA level, this is somewhat lower than the 0.5 value observed in a $\phi$X174 experiment (*Wichman et al.*, *1999*), but greater than 2–3% values observed in *E. coli* experiments (*Woods et al.*, *2006*; *Tenaillon et al.*, *2012*). The increased level of parallelism at the regulatory/codon level is qualitatively similar, if less dramatic, than the findings in *E. coli* where parallelism increases an order of magnitude at the gene/operon level. (Because there are only 11 genes in the bacteriophage, it is not all that meaningful to ask how often lines share changes in the same gene.)

To make our analysis more precise, we define between-well parallelism as the probability that a mutation observed in one well was also be found in a second well. We calculated observed parallelism for all pairs of wells initiated from the same background, denoted $i$ and $j$, by determining the proportion of mutations from $j$ found in $i$, the proportion of mutations from $i$ found in $j$, and averaging. At the nucleotide level, mean within-background parallelism ranged from 0.15 in F421 to 0.46 in F416, with an average of all nine backgrounds of 0.28 (SE = 0.03; Fig. 5A). The reason F416 shows such high parallelism is that reversions occurred in 7 of 8 cases (Fig. 4B) and, in each case, mutation 2534gT also arose. No other background experienced more than 3 reversions. When reversions are removed from the dataset, parallelism at F416 drops to 0.38 and the global mean drops just slightly to 0.27. We conducted a randomization test that accounts for the nonindependence of pairwise comparisons and found that, with reversions excluded, there is no evidence that parallelism differs among backgrounds ($p \approx 0.58$).

We have already shown that the D-promoter mutations are found in most wells (Fig. 4A). To quantify their contribution to parallelism, we removed the D-promoter sites and reanalyzed the data. Within background parallelism falls from an average of 0.28 to 0.13. Thus, slightly more than half of the nucleotide level parallelism comes from the D-promoter mutations.

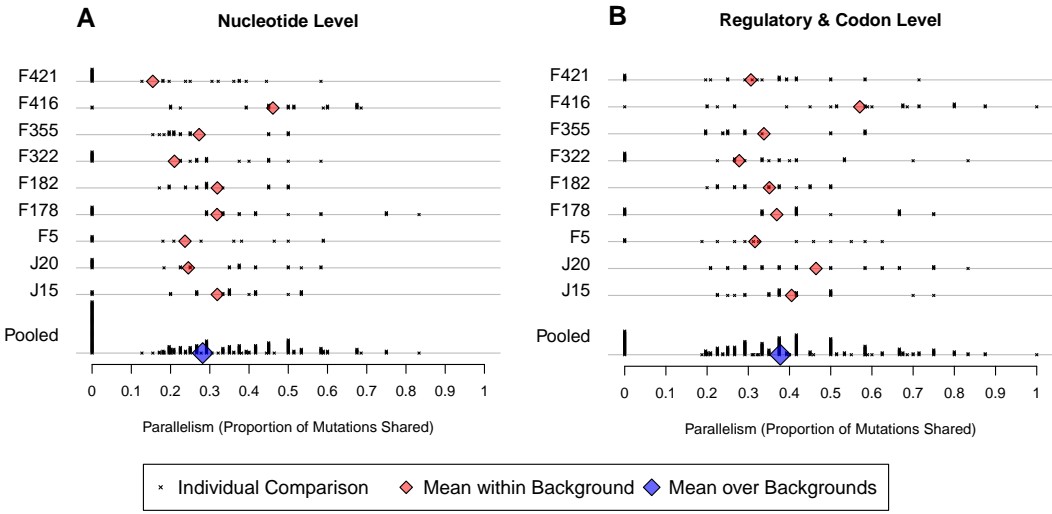

**Figure 5  Amount of parallelism between pairs of wells within backgrounds.** Parallelism is defined as the mean of the proportion of mutations in one well shared with the other and the reciprocal comparison. Note that individual pairwise comparisons (x symbols) with same value are stacked vertically, creating the bars. (A) Mutations are shared only when they match at the nucleotide level. (B) Mutations match when they occur at the same codon or in the same regulatory element. Backgrounds are indicated on the $y$-axis; only within-background comparisons are made. Background F416 has elevated parallelism in large part because it experienced much higher reversion (Fig. 4). Once reversions are removed, a randomization test showed no evidence for differences in parallelism between backgrounds at the nucleotide level ($p \approx 0.58$) nor the regulatory/codon level ($p \approx 0.56$).

A broader view of parallelism is to categorize mutations as the same when they occur in the same regulatory element or codon. When we do this, parallelism increases from an overall mean of 0.28 to 0.38 (Fig. 5B). The randomization test indicates that F416 is different from the other backgrounds ($p \approx 0.013$), but the other eight backgrounds show no evidence of having different levels of regulatory-level parallelism ($p \approx 0.228$). Removing reversions from the dataset has a minimal effect on mean parallelism (0.378 falls to 0.376). When F416 is removed (but reversions are left in), mean parallelism falls from 0.38 to 0.35. When the D-promoter mutations are again excluded from the analysis, the overall mean of this broad parallelism falls to 0.17. We regard 38% to be our single best characterization of parallelism in this experiment, with slightly more than half of this owing to parallel changes in the D-promoter.

## Epistasis

Epistasis refers to the way in which the genetic background influences a mutation's fitness effect. Parallelism is increased when beneficial mutations have approximately the same effect on all genetic backgrounds and it will decrease when the fitness effects (including their sign) depend strongly on the genetic background. By initiating our replicate adaptations from nine different backgrounds, our experiment was explicitly designed to test how epistasis influences where mutations arise and, thereby, the amount of parallel evolution in our system.

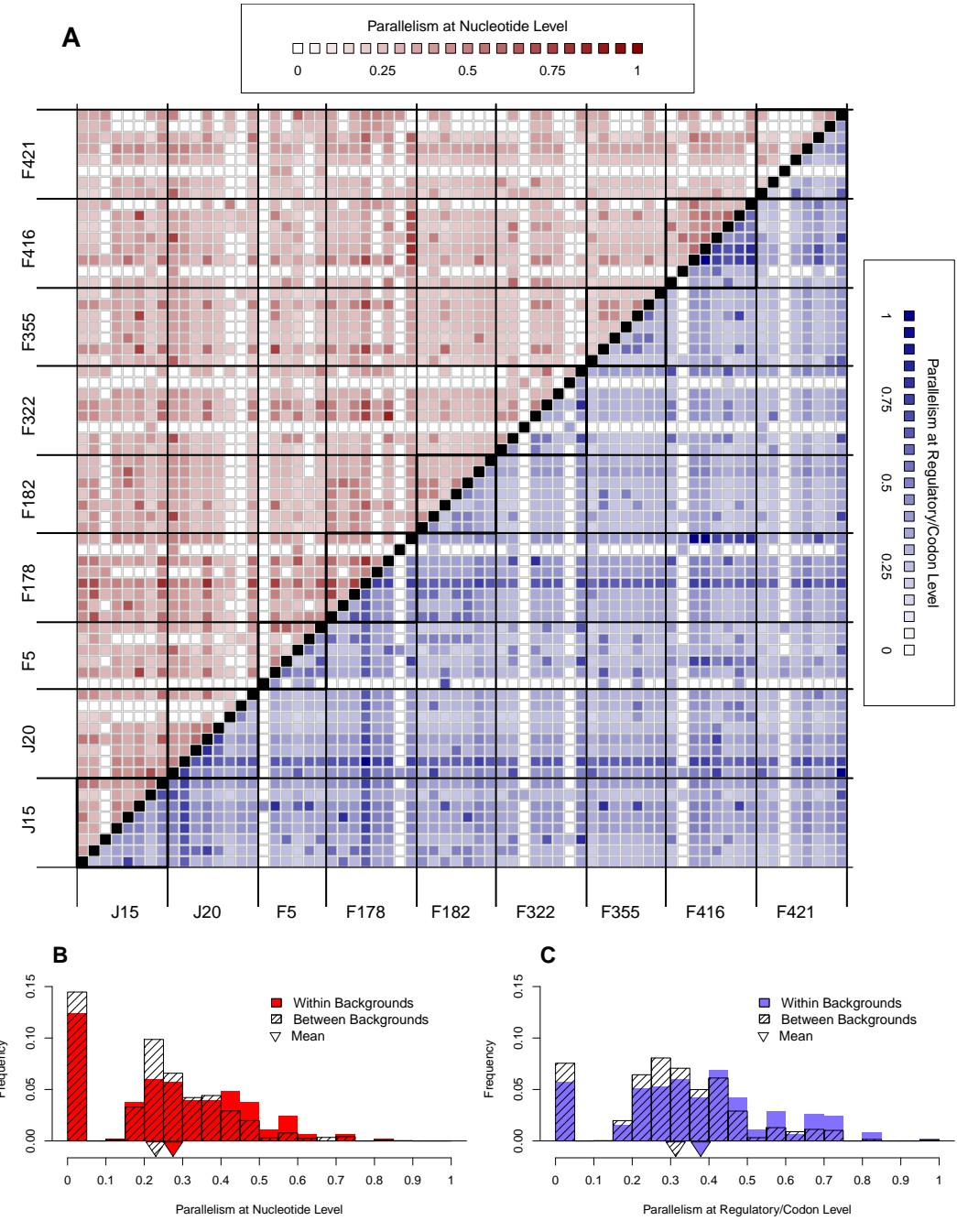

**Figure 6** **Parallelism within backgrounds is only slightly higher than parallelism between backgrounds.** (A) Parallelism between all pairs of wells from low (light) to high (dark). Above the diagonal in red is parallelism at nucleotide level; below the diagonal in blue is parallelism at the regulatory/codon level. Squares along diagonal are within background comparisons, all others are between backgrounds. Notice that within-background quadrants are not noticeably darker than between-background quadrants. (B–C) Histograms comparing within- vs. between-background parallelism pooled across backgrounds at the nucleotide (B) and regulatory/codon (C) level. The histograms reveal that parallelism is, on average, slightly higher within backgrounds ($p < 0.01$).

### Effect of Background

We have already seen that reversion rate is significantly different among backgrounds (Fig. 4B). In this section, we begin looking for background epistasis by asking whether there is a difference in parallelism within- vs. between-backgrounds once reversion are removed. Figure 6A shows a heatmap of all pairwise well comparisons at both nucleotide (red) and regulatory/codon (blue) levels. Reversions have been removed in this analysis. Notice that visually the within-background comparisons along the diagonal are not much different than those off the diaganol by either metric of parallelism. This suggests that parallelism is similar within and between backgrounds. However, when we pool all observations within vs. between, we find that the distribution of within-background parallelism is slightly greater than between (Fig. 6B). Thus, parallel mutations are slightly more likely to occur within wells of the same background than of different backgrounds. A formal randomization test accounting for nonindependence of the observations confirms that the difference in means is significant ($p < 0.01$). The implication of the within- vs. between-background analysis of Fig. 6 is that while the occurrence of mutations is highly stochastic, genetic background does exert a slight but significant influence over where mutations appear.

A second, more powerful, way to assess the role of background epistasis is to ask whether background affects the individual probability of a mutation arising. More formally, does a model where the probability of a mutation occurring depends on background fit the data significantly better than a null model where background does not matter? Given the parallelism results just presented, it is not surprising that in a likelihood ratio test (LRT) of these competing hypotheses the null can be rejected ($p < 0.001$). The main advantage of the LRT approach is that the disparity between the alternative and null hypotheses (i.e., the difference in the log-likelihoods, or $\Delta lnL$), can be broken down by mutation. Figure 7 shows the number of times each mutation appears on each background and, to the right, $\Delta lnL$ for each mutation. The larger bars are the mutations most responsible for the association of mutation with background. We removed mutations in a cumulative manner starting with the largest $\Delta lnL$ value and reran the LRT until we failed to reject. This revealed that just four of the mutations (red bars in the figure) are principally responsible for the significant relationship: 2534gT, 2397aG, 2332aG, and 2100tC. Thus, we again find that background epistasis exists, but the effects are substantial for only a small minority of the mutations.

### Epistasis between de novo mutations

Another type of epistasis involves combinations of new mutations. For our data, the signal for this is when mutations appear together significantly more or less often then we would expect by chance. We conducted an analysis on mutation pairs, asking if they tend to co-occur or avoid each other. The results, summarized in Fig. 8, revealed two interesting patterns. First, despite finding D-promoter mutations in 62 of 68 (91%) wells and in 273 of 338 (81%) isolates, no two D-promoter mutation are ever found together in the same isolate (Fig. 8). The significance of this is seen clearly in the figure by focusing on the *D-pro block* in the bottom row which represents the combined set of all the D-promoter mutations. A bootstrap randomization indicated we expect D-promoter mutations to co-occur 19

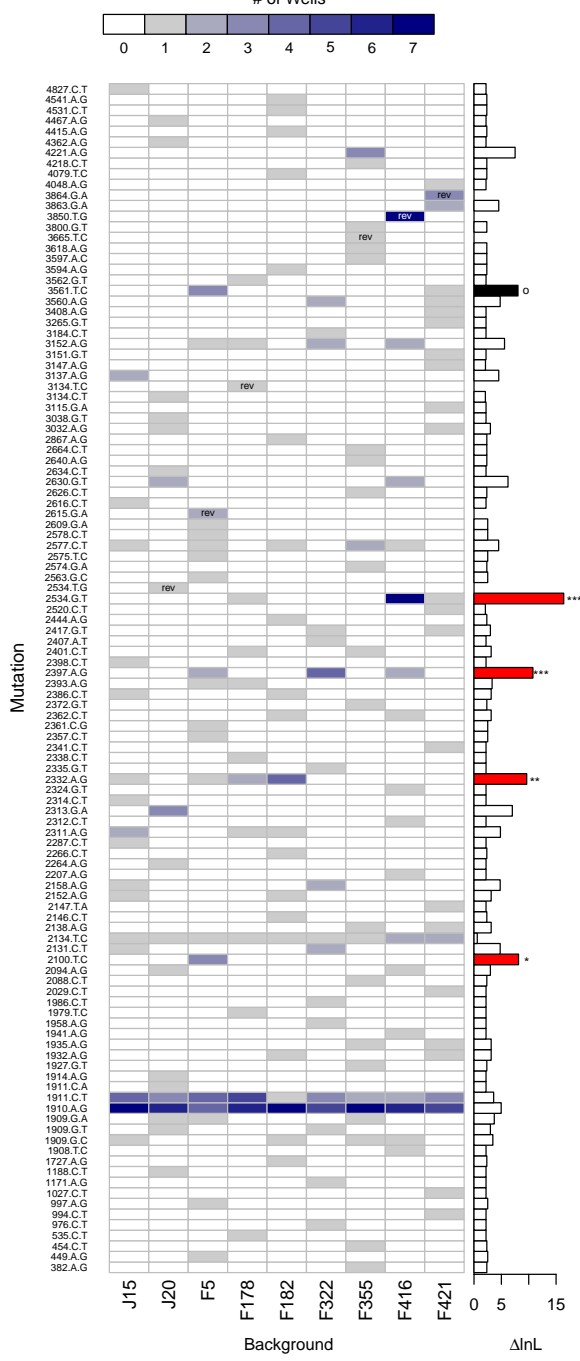

**Figure 7 Evidence of background epistasis is significant but rare.** The main plot shows the number of wells each mutation appears in with a heatmap scale above the plot. Within cells, 'rev' indicates the mutation is a reversion and is excluded from statistical testing. The barplot to the right gives the difference in log-likelihood ($\Delta lnL$) between the null (no epistasis) and the alterantive (epistasis) model for each mutation. Globally, the no-background effect null is rejected ($p < 0.001$). To determine which mutations are significant, they were removed from the dataset from the largest downward in a cumulative fashion. Removal of the mutations with red bars still resulted in significant results (\*\*\*$p < 0.001$, \*\*$p < 0.01$, \*$p < 0.05$). Removal of the red mutations plus the mutation in black results in a non-significant result ($p \approx 0.104$).

times; observing 0 is highly nonrandom ($p < 0.01$). Second, several other mutations are never found with a D-promoter mutation. Most notably, mutation 2134tC is found in 11 different wells and is expected to co-occur with a D-promoter mutation 7 times, but it never does. Mutations 2094aG, 2131cT and 2158aG are also never observed with a D-promoter mutation, but these mutations are rare enough that the observed repulsion is not significant. Finally, the analysis shows several significantly positive associations that suggest synergistic epistasis: 2534gT co-occurs with D-promoter mutations and with 2630gT, and 2577cT occurs with the D-promoter mutation 1910aG more often than expected by chance.

## Parallel evolution at the phenotypic level

The most striking result from analysis of epistasis is that while D-promoter mutations are pervasive, they never co-occur in the same isolate, and a number of other mutations are also mutually exclusive of the D-promoter mutations and each other. One parsimonious explanation of this pattern is that these mutually exclusive mutations have closely related phenotypic effects and that having two such mutations is either neutral or deleterious. We now show that the phenotype involves delay of lysis. We postulate that selection favored lysis delay because, under high MOI conditions, lysing early has a diminished benefit (finding another host cell is improbable) compared to the advantage of exploiting the current host more effectively, i.e., producing more progeny.

The first evidence of what selection is doing here comes from experimental evolution on $\phi X174$. In chemostat adaptation experiments where high MOI conditions prevail, D-promoter mutations are commonly observed (*Wichman et al.*, *1999*; *Wichman, Millstein & Bull*, *2005*). *Brown et al. (2010)* studied four D-promoter mutations and found that they all reduce gene D/E mRNA transcript levels on the order of 75–80%. Two of the mutations studied there are the same as the two most commonly observed in our data: 1910aG and 1911cT. Since the lysis protein E is encoded in this transcript, protein E levels are presumably reduced by these mutations. We therefore suspected that our D-promoter mutations are down-regulating the lysis protein in our experiment and thereby delaying lysis.

This led us to examine the 65 isolates that lack a D-promoter mutations. Might these be achieving similar ends by different means? We found that 56 of these 65 isolates (86%) have mutations that could reasonably affect the E protein in that the mutations (i) are either just upstream of the E protein coding region or in the gene itself and (ii) are never found in an isolate with a D-promoter mutation (46 of 56 have the aforementioned mutations 2094aG (5 isolates), 2131cT (4 isolates), 2134cT (26 isolates) and 2158aG (11 isolates) while the other 10 have mutations found in just one well: 2147tA (2 isolates), 2335gT (1 isolate), 2361cG (5 isolates), 2372gT (1 isolate) and 2407aT (1 isolate)). Thus, only 9 of the total 338 isolates (2.7%) had neither a mutation in the D-promoter nor a change that might reasonably effect the E protein.

These observations led us to do lysis assays on a subset of mutations we hypothesized were delaying lysis: 1910aG and 1911cT (the two common D-promoter mutations found in 182 and 62 isolates respectively), 2131cT (found in 4 isolates) and 2134cT (found in 26 isolates). These mutations were selected in part because of their abundance (representing 274 of 338 isolates, or 81%) and because we already had sequenced isolates with and without

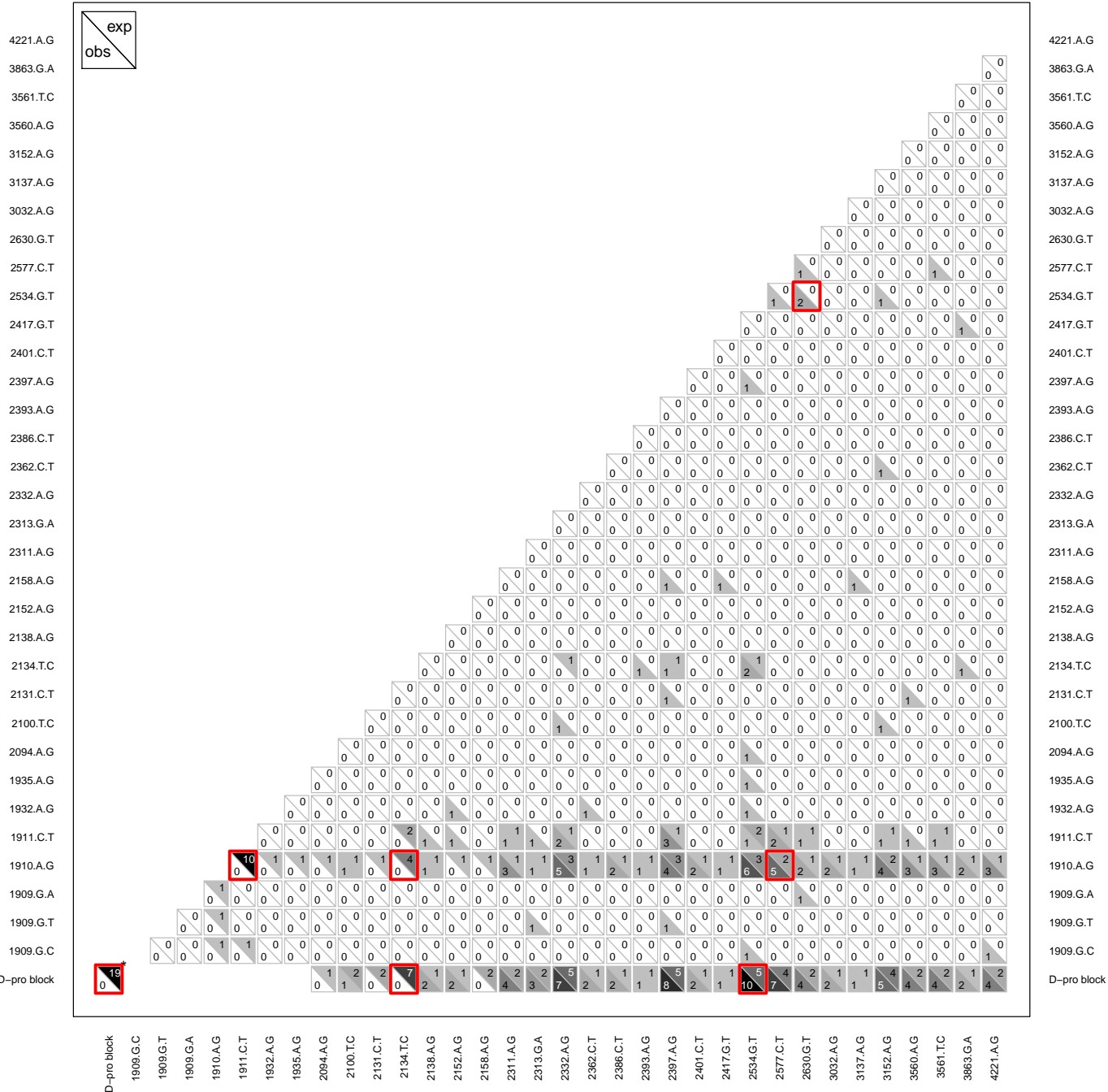

**Figure 8** **Pattern of observed vs. expected co-occurences.** Most importantly, the plot reveals that D-promoter mutations never co-occur nor do they occur with several other mutations including 2134cT, 2094aG, 2131tC, and 2158aG. Each cell shows the observed count in the lower left vs. the expected count in the upper right for that comparison. Larger counts are shaded more darkly to show the overall pattern and significant differences ($p < 0.05$) are highlighted in red. The D-promoter mutations are at sites 1909–1935 (first 7 mutations) and the *D-pro block* is this collection of mutations. *The D-pro block comparison with itself shows the total number of observed and expected wells where two different D-promoter mutations co-occur in the same isolate; all other comparisons with D-pro block are observed and expected co-occurences with any D-promoter mutation. Singletons and reversions have been removed from the figure and expected counts have been rounded to nearest integer. Expected counts are based on a bootstrap randomization procedure (see methods).

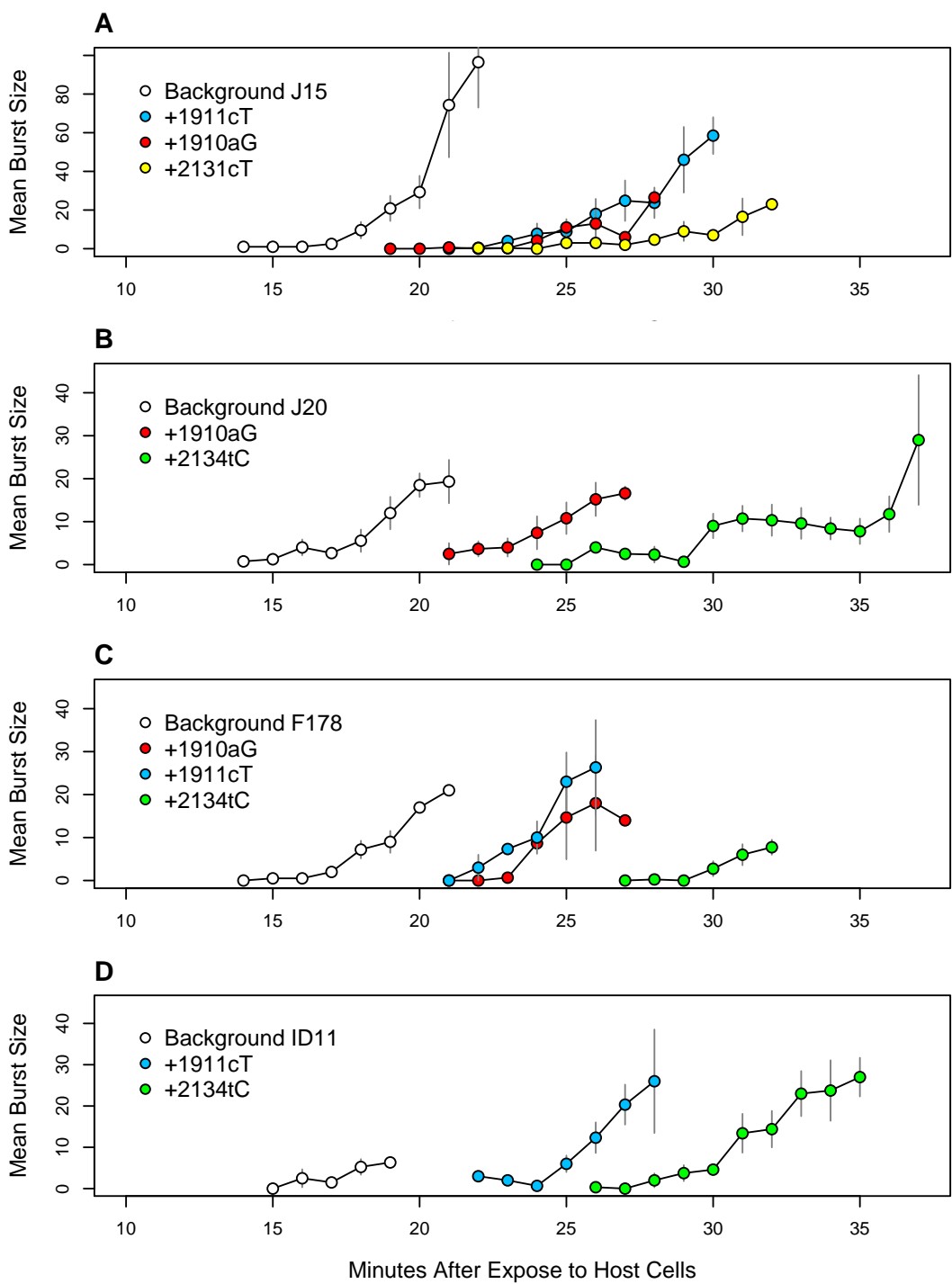

**Figure 9** **Time to lysis is delayed by the assayed mutations across backgrounds.** Each panel presents a different genetic background: (A) J15, (B) J20, (C) F178, (D) ID11 wild type. Plotted are the means and standard errors of all observed burst counts for each genotype at each time point.

each mutation. For three of them (all but 2131cT) we had isolates allowing the with/without comparison to be made on multiple backgrounds. This let us assess phenotypic effect of each mutation across multiple genetic backgrounds. The results, presented in Fig. 9, confirm our hypothesis. The D-promoter mutations delay lysis 5–8 min depending on the mutation and the background. Mutations 2131cT and 2134cT delay lysis on the order of 12–13 min. The patterns are very similar across background suggesting that this phenotypic effect is not very background dependent. Thus, we have strong evidence that over 80% of the isolates are delaying lysis and, based on the arguments above, there is reason to suspect that most of the remaining isolates have delayed lysis as well. At the well level, at least 64 of the 68 wells ($\geq$94%) have mutations that delay lysis or are strongly suspected of doing so.

Given this dramatic delay in lysis time, we were interested in what molecular mechanism(s) underlie the delay. With the D-promoter mutations, we hypothesized that the mutations change the binding affinity of the RNA polymerase which alters expression of the lysis protein, protein E (*Brewster, Jones & Phillips*, *2012*; *Kinney et al.*, *2010*). For mutations downstream of the D-promoter within the D/E genes, we hypothesized that delay is driven by other means of delaying protein E expression. We considered the possibility that E is down-regulated by mutation to rare codons (*Sharp, Emery & Zeng*, *2010*), ribosomal pausing at mutationally introduced Shine-Delgarno-like sequences (*Li, Oh & Weissman*, *2012*) or mutationally altered transcript stability (*Agashe et al.*, *2013*). As we detail in the Supplemental Information, our analyses failed to find strong support for any of these hypotheses by themselves. We continue to suspect that lysis delay is driven by change in protein E expression. We speculate that our inability to uncover drivers may come from having relatively few mutations, from using models (as opposed to molecular data) for exploring the mechanisms and the possibility that multiple mechanisms may be involved across and even within individual mutations.

## CONCLUSIONS

Our inability to make precise predictions about evolution, even in the laboratory, reveals how much is yet unknown about the process. The study of parallelism is useful, in part because it illuminates how inherently stochastic evolution is and, therefore, how good our predictions may become as knowledge improves. Experimental evolution remains in the very early stages of tackling the problem. A number of studies have sought to gather insight by taking one adaptive outcome (i.e., a collection of beneficial mutations), engineering all the possible pathways to it, and asking how viable is each pathway (*Weinreich, Watson & Chao*, *2005*; *Poelwijk et al.*, *2007*; *Chou et al.*, *2011*; *Khan et al.*, *2011*). The advantage of this approach is that it reveals a small region of the adaptive landscape in great detail; the problem is that it generally leaves the vast majority of the evolutionarily relevant landscape unexamined. Relatively few studies have attempted to study the broader fitness landscape and ask about the multiplicity of trajectories evolution might take (*Wichman et al.*, *1999*; *Woods et al.*, *2006*; *Tenaillon et al.*, *2012*; *Lee & Marx*, *2013*; *Lang et al.*, *2013*; *Frenkel, Good & Desai*, *2014*).

Here we sampled and characterized the range of potential short-term trajectories in a bacteriophage by allowing eight replicate lines with each of nine first-step mutational

backgrounds to adapt for 100 passages. Our results provide both informative details and, stepping back, a simple storyline. The details can be summarized in several succinct points. (1) Our populations generally displayed complex interference dynamics with many mutations arising and competing simultaneously. (2) A lot of variation–and adaptive variation–arose: 112 mutations and 281 mutational events occurred with at least 40% of the mutations and 77% of the events being adaptive. (3) The variation was concentrated: over 80% of the mutational events arose in the D-promoter, the J-terminator and genes D, E and F. (4) Mutational appearance was highly skewed: a few mutations in the D-promoter arose a lot, a few mutations arose a modest number of times and most mutations arose between once and just a few times. (5) Overall, parallelism was modest: wells starting with the same background shared, on average, 27% of their mutations at the DNA level and 38% of their changes at the broader codon/regulatory level. About half of this parallelism came from changes in the D-promoter. (6) Epistasis was present, but it was largely driven by a handful of interactions: a few mutations arose at elevated rates on certain backgrounds, several mutations tended to co-occur and, most importantly, D-promoter mutations and a number of other mutations within the gene D/E transcript never co-occurred. The likely reason for this is that the mutations in repulsion are all doing the same thing: delaying lysis time.

Thus, at least one phenotypic change–delay of lysis–was observed in parallel over most of our replicate lines. 94% of the wells and over 85% of the isolates had mutations that we know or strongly suspect delay lysis. This leads us to hypothesize that under our high MOI conditions, selection favored phage delaying the pursuit of a fresh host to better exploit the host they are in (i.e., love the one the one you're with). In some ways our findings echo those of other recent studies that have looked at replicate adaptation and shown or implicated that parallelism is much higher at the phenotypic level than the genetic level (*Woods et al.*, *2006*; *Saxer, Doebeli & Travisano*, *2010*; *Lee & Marx*, *2013*; *Tenaillon et al.*, *2012*). Our work, like theirs, reinforces the view that answering questions about the range of possible adaptive trajectories, parallelism, and the predictability of evolution requires attention to the many biological levels where the process of adaptation plays out.

### Funding
This work was supported by National Institutes of Health grants R01 GM076040-01. Genomic and computational resources were supported by P30 GM103324. The funders had no role in study design, data collection and analysis, decision to publish, or preparation of the manuscript.

### Grant Disclosures
The following grant information was disclosed by the authors:
National Institutes of Health: R01 GM076040-01.
Genomic and computational resources: P30 GM103324.

### Competing Interests
The authors declare there are no competing interests.

## Author Contributions

- Craig R. Miller conceived and designed the experiments, performed the experiments, analyzed the data, wrote the paper, prepared figures and/or tables, reviewed drafts of the paper.
- Anna C. Nagel conceived and designed the experiments, performed the experiments, reviewed drafts of the paper.
- LuAnn Scott conceived and designed the experiments, performed the experiments, contributed reagents/materials/analysis tools, reviewed drafts of the paper.
- Matt Settles conceived and designed the experiments, analyzed the data, contributed reagents/materials/analysis tools, prepared figures and/or tables, reviewed drafts of the paper.
- Paul Joyce conceived and designed the experiments, reviewed drafts of the paper.
- Holly A. Wichman conceived and designed the experiments, contributed reagents/materials/analysis tools, wrote the paper, reviewed drafts of the paper.

## DNA Deposition

The following information was supplied regarding the deposition of DNA sequences:

The 338 isolate sequences used in this paper have been deposited as a PopSet in GenBank under accession numbers KX266300–KX266637. Each sequence corresponds to an endpoint isolate with the name uniquely identifying the isolate. Names follow the convention, Plate_Well_Background_Isolate#.

## Data Availability

The raw data has been supplied as Supplemental Dataset.

## Supplemental Information

Supplemental information for this article can be found online at http://dx.doi.org/10.7717/peerj.2227#supplemental-information.

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
