# Peer review of "Love the one you’re with: replicate viral adaptations converge on the same phenotypic change"

_PeerJ, doi:10.7717/peerj.2227_

## Round 0.1 · original submission · Minor Revisions

· Academic Editor

Minor Revisions

I really enjoyed the structure and content of this manuscript, and commend the authors for their extensive and well-organized supplementary data. Both reviewers make a number of suggestions that should improve the manuscript after minor revision.

Reviewer 1 ·

Basic reporting

There are several minor typos throughout (e.g. "lead" instead of "led", "epistatsis", "seciton", "bare" instead of "bear"), please give careful read for these.

In the abstract I was confused by the distinction between "112 de novo mutations" and "281 mutational events". It would be worth making clear that you mean a total of 281 mutational events, which involve 112 unique changes.

Experimental design

The experimental design is appropriate for the questions being addressed.

Validity of the findings

In the discussion of clonal interference/multiple mutations on lines 226-230: We would sometimes expect to see a branching tree even in the classic Gerrish+Lenski model of clonal interference (when one mutation was being interfered with by another). The authors are right that the prevalence of branching trees suggests that concurrent mutations are common, but the presence of a branching tree by itself is not definitive proof in any single case. A slight rewording here would make this more clear.

What would be the null expectations for nonsynonymous to synonymous mutations in a totally neutral situation (This is not obvious, especially given overlapping reading frames)?

I'm not convinced by the approach used to identify neutral versus beneficial mutations (supplemental info section 1.4). The authors are arguing that if there is a single mutation that reaches substantial frequency without any other identified mutation, it must be beneficial. Similarly, if a mutation spawns two or more descendant lineages with different mutations on them, they claim it must be beneficial. The first of these tests could fail if there are additional mutations that are missed by the sequencing (presumably the authors cannot be sure they identify all mutation events). The second of these tests could fail for the same reason, or if one of the descendant lineages is a low-frequency neutral/deleterious mutation that is lucky to be sampled. To be clear, both of these caveats are unlikely to explain the majority of the cases, so the general conclusions the authors are drawing are almost certainly correct. However, quoting specific percentages as confident lower bounds may not be wise.

Reviewer 2 ·

Basic reporting

This article meets all of PeerJ's basic reporting requirements. The article is clearly written. All of the raw data, including the scripts used for data analysis are provided in the article or in supplementary materials.

Experimental design

This experiment addresses an important question in the field of evolutionary biology regarding the extent to which adaptation is stochastic vs deterministic. The authors do an excellent job of analyzing the prevalence of parallel evolution in their experimental phage lineages at several different levels of biological complexity. They also do a thorough job of addressing the role of epistasis in their results. The finding that complex interference dynamics were dominant in most of the wells also adds to the body of results describing mutational dynamics in experimentally evolved populations.

I am curious to know why competition assays were used to determine the selection coefficients of each of the original nine first-step mutations under the experiment's high MOI conditions, instead of following the assay method originally used to determine the fitness of each of these mutants by Rokyta et al. (2005), where the fitness, measured as growth rate, was determined independently for each isolate.

Validity of the findings

The findings appear to be valid.

There are a couple of spots in the results section under the heading "Parallel Evolution by Well" where I think the phrasing of the calculations of the rate of parallelism is a bit confusing. If I am reading this section correctly, all of these results are calculations for pairs of wells within each of the nine backgrounds, but the way the results are described makes it sound at times as though the calculations used random pairs of wells across all backgrounds. Specifically, I think the second sentence in the first paragraph could read, "We now change the conditioning and ask, for two independent wells *from within the same background*, what proportion of the mutations observed in them are shared." Again, in the third paragraph, which reads "Parallelism falls to an across background average of 0.13," I think the authors actually mean that parallelism within each set of wells begun from the same isolate falls to an across background average of 0.13. But perhaps I am just misreading the results.

In the next to last paragraph, the authors state that 112 mutations and 338 mutational events occurred in the experimental populations. I believe this should actually be 281 mutational events. 338 is elsewhere listed as the number of isolates sequenced.

---

## Round 0.2 · accepted · Accept

· Academic Editor

Accept

They authors have done an excellent job in both the study and in replying to the reviewers' comments. I really appreciate the detailed supplementary methods and the overall level of care that went into this work.

Reviewer 1 ·

Basic reporting

No Comments

Experimental design

No comments

Validity of the findings

I appreciate the authors' care in responding to my concerns from the previous version of the manuscript. I am satisfied with the changes they have made in response to some of my comments, and in the other cases I'm convinced by their rebuttal arguments (most importantly, I'm used to thinking about sequencing studies in microbes where alignment artifacts and coverage variation mean that one can never be sure that all mutations have been identified, but I appreciate the authors' point that their case here is quite different).

Reviewer 2 ·

Basic reporting

This article still meets all of PeerJ's basical reporting guidelines.

Experimental design

The authors have adequately addressed my question regarding their use of competition assays rather than flask assays to determine selection coefficients of the starting mutations under high MOI conditions.

Validity of the findings

The authors have addressed the two points in the article where I thought their wording may have been confusing.

Additional comments

All of my concerns from the first review have been adequately addressed, and I have no further questions or concerns. I think the article is excellent and should be accepted as is.